# A Compact Particle Detector for Space-Based Applications: Development of a Low-Energy Module (LEM) for the NUSES Space Mission

**Riccardo Nicolaidis** [1,2,*,†] , **Francesco Nozzoli** [1,2,†] , **Giancarlo Pepponi** [3,†]
**and on behalf of the NUSES Collaboration** [‡]

1   Physics Department, University of Trento, Via Sommarive 14, 38123 Trento, Italy; francesco.nozzoli@unitn.it
2   INFN-Trento Institute of Fundamental Physics and Applications, Via Sommarive 14, 38123 Trento, Italy
3   Fondazione Bruno Kessler, Via Sommarive 18, 38123 Trento, Italy; pepponi@fbk.eu
*   Correspondence: riccardo.nicolaidis@unitn.it
†   These authors contributed equally to this work.
‡   Collaborators of the NUSES Collaboration are indicated in Supplementary Materials.

**Abstract:** NUSES is a planned space mission aiming to test new observational and technological approaches related to the study of relatively low-energy cosmic rays, gamma rays, and high-energy astrophysical neutrinos. Two scientific payloads will be hosted onboard the NUSES space mission: Terzina and Zirè. Terzina will be an optical telescope readout by SiPM arrays, for the detection and study of Cerenkov light emitted by Extensive Air Showers generated by high-energy cosmic rays and neutrinos in the atmosphere. Zirè will focus on the detection of protons and electrons up to a few hundred MeV and to 0.1–10 MeV photons and will include the Low Energy Module (LEM). The LEM will be a particle spectrometer devoted to the observation of fluxes of relatively low-energy electrons in the 0.1–7-MeV range and protons in the 3–50 MeV range along the Low Earth Orbit (LEO) followed by the hosting platform. The detection of Particle Bursts (PBs) in this Physics channel of interest could give new insight into the understanding of complex phenomena such as eventual correlations between seismic events or volcanic activity with the collective motion of particles in the plasma populating van Allen belts. With its compact sizes and limited acceptance, the LEM will allow the exploration of hostile environments such as the South Atlantic Anomaly (SAA) and the inner Van Allen Belt, in which the anticipated electron fluxes are on the order of $10^6$ to $10^7$ electrons per square centimeter per steradian per second. Concerning the vast literature of space-based particle spectrometers, the innovative aspect of the LEM resides in its compactness, within $10 \times 10 \times 10$ cm$^3$, and in its "active collimation" approach dealing with the problem of multiple scattering at these very relatively low energies. In this work, the geometry of the detector, its detection concept, its operation modes, and the hardware adopted will be presented. Some preliminary results from the Monte Carlo simulation (Geant4) will be shown.

**Keywords:** low energy module; NUSES; particle bursts; silicon detectors; PIPS; cosmic rays; particle identification; Δ*E-E* telescope

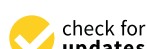



## 1. Introduction: The NUSES Space Mission

NUSES is a space mission aimed at testing innovative approaches for studying cosmic rays, gamma rays, and astrophysical neutrinos. The satellite will host two payloads, named Terzina [1,2] and Zirè [3–7]. Terzina will be a pathfinder for future missions devoted to observing Ultra High Energy Cosmic Rays (UHECRs) and neutrino astronomy using space-based instruments [8–11].

Zirè is a particle detector that tests novel instruments for the detection of γ-rays while monitoring fluxes of charged particles, such as electrons, protons, and light nuclei with kinetic energy from a few to hundreds MeV. The detector's primary goal is to count

the trapped particles precipitating out of the Van Allen Belts (VABs) and look for any anomalies that might arise in the vicinity of tectonic events, including earthquakes or lithosphere-volcanic eruptions.

Monitoring solar activity and its cyclical cycle, which lasts roughly 11 years, is another crucial science goal of Zirè. Monitoring the incidence of phenomena like Solar Flares (SFs) or Coronal Mass Ejections (CMEs) during solar maximum is especially helpful [12]. The Zirè instrument will allow an online monitoring of the magnetospheric environment, useful for space weather characterizations Moreover, the study of energetic particles in the magnetosphere will advance our knowledge of the acceleration mechanisms at work during those occurrences. In conclusion, detecting photons with energy up to tens of MeV is the other science goal. This enables the investigation of some of the most intense and violent occurrences in astrophysics, known as Gamma Ray Bursts (GRBs) [13,14], which are fast and powerful gamma-ray pulses originating from extremely distant sources.

An additional detector, the Zirè-Low Energy Module (LEM), will cooperate with Zirè. The LEM is going to be inserted into the outer structure of the NUSES bus. Figure 1a shows the NUSES satellite. The on-board payloads are labeled at the edges of the figure. The LEM sub-detector has been designed to fit within a $10 \times 10 \times 10$ cm$^3$ volume. The LEM's goal will be to accomplish event-based particle identification (PID) for particles with relatively low kinetic energy, such as sub-MeV electrons and MeV protons.

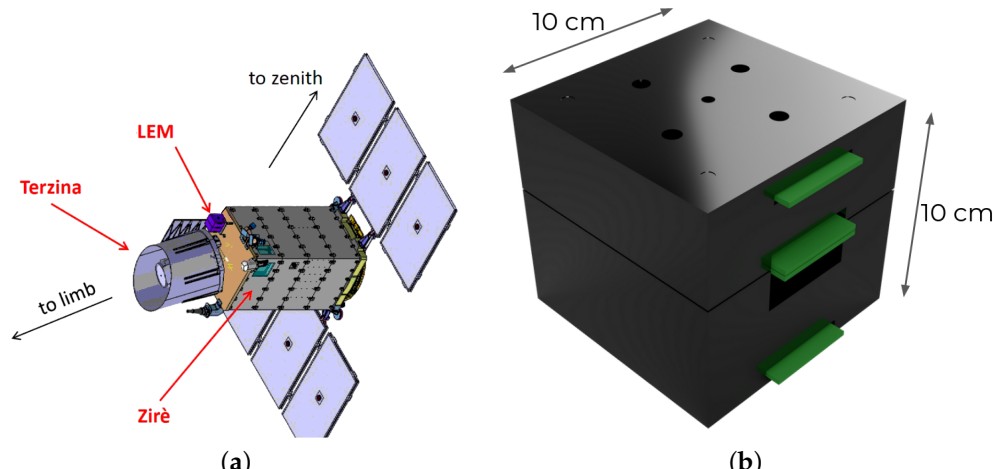

(**a**)　　　　　　　　　　　　　　　　　　　　　　(**b**)

**Figure 1.** (**a**) The NUSES platform can be visualized in 3D. The Terzina detector faces the Earth's limb and measures fluorescence light from Extensive Air Showers (EAS) or Upgoing Air Showers (UAS) that are caused by the decay of the neutrino $\tau$. The Zirè-LEM is shown as a small purple box. The detector is positioned outside the satellite's tray and is pointing towards the zenith. Lastly, Zirè is located inside the tray of the satellite and has three windows facing external space. (**b**) A visual representation of the Zirè-LEM detector. The 8 mm thick aluminum shield is depicted in the picture as a dark surface. Its goal is to reduce the occupancy of the veto scintillators, absorbing a large fraction of sub-MeV electrons. The particles that are the target are those whose incident directions allow them to enter the detector through the five holes shown in the picture.

## 2. The Need for a Low Energy Module

One of NUSES's key missions, as previously stated, is to monitor particle precipitation from the Van Allen Belts (VABs) and investigate eventual correlations to seismic activity, all while validating models for Lithosphere–Atmosphere–Ionosphere–Magnetosphere interactions. Several processes can result in the release of trapped charged particles from radiation belts. While our understanding of these processes is still limited, electromagnetic fluctuations within the radiation belt are widely assumed to be a significant factor. Geomagnetic/solar storms, thunderstorms, but also human-generated electromagnetic emissions, and seismic events can all cause these electromagnetic fluctuations. As highlighted in [15],

measuring the fluxes of trapped charged particles can improve our understanding of the connection between the lithosphere, atmosphere, ionosphere, and magnetosphere. As pointed out in [16], there is statistical evidence indicating a temporal correlation between particle precipitation from the Van Allen Belts and major seismic events. These findings stimulate interest in more accurate measurements of electron fluxes with energies spanning in the range of 0.1–7 MeV, which could be a candidate channel for identifying hypothetical seismic precursors.

As a result, a Low-Energy Module (LEM) has been included in the design of the NUSES satellite in order to expand the observed energy window by Zirè. The Low Energy Module will be a small spectrometer with a volume of $10 \times 10 \times 10$ cm$^3$ and mass less than 2 kg (Figure 1b for the volume envelope of the LEM), capable of measuring the kinetic energy, arrival direction of low energy charged particles down to 0.1 MeV for electrons.

The primary goal of this detector is to observe the magnetosphere and ionosphere surroundings. Furthermore, the LEM instrument will investigate particle composition in the challenging conditions of the South Atlantic Anomaly (SAA), to quantify the isotopic ratios of H and He and potentially determine the proportion of heavier nuclei.

## 3. Geometry and Detection Concept of the LEM

The direction of a particle in a particle spectrometer, such as the Zirè, is usually established using tracking techniques. When dealing with relatively low-energy particles, however, the conventional tracking method is not feasible due to significant multiple scattering within the first sensitive element of the direction detector. A collimation technique, as discussed in [17], is required to determine the arrival direction of relatively low-energy particles. This approach requires utilizing a well-constructed passive shield with adequate thickness for blocking energetic particles coming from "unidentified" or random directions.

The passive collimator makes it simpler to detect particles arriving from "accepted" directions. To avoid the need for large and heavy passive protections, the LEM spectrometer employs a technique known as "active collimation". This method employs shaped plastic scintillators as ACD, successfully distinguishing particles that pass through a relatively lightweight passive shield.

The LEM features a 0.8-cm thick aluminum barrier to maintain bearable occupancy levels for the veto detectors in places such as the South Atlantic Anomaly (SAA) and the inner radiation belt adjacent to the poles, enabling consistent particle composition observations in those regions.

The LEM's particle identification capabilities rely on the long-established $\Delta E$-$E$ spectrometric method, which is discussed in references [18–20]. The approach employs five couples of silicon detectors (Passivated Implanted Planar Silicon or PIPS) positioned in a telescopic arrangement. The PIPS detector has a typical resolution of about 10 keV.

In Figure 2, a representation of the detector shows the components and its preliminary assembly. On the other hand, in Figure 3, a schematic view of the instruments explains its adopted detection scheme. A particle, when approaching the detector, can enter the instrument through the holes in the top aluminum structure, preventing the detection by the perforated top ACD (made of plastic scintillator). Depending on the particle's direction, charge and kinetic energy are determined by one of the five $\Delta E$-$E$ spectrometers. Each spectrometer consists of a thinner silicon detector (100 μm thick) placed on top of a thicker silicon detector (300 μm thick).

To improve the LEM's particle detection capabilities across a wider energy range, a calorimeter is placed beneath the PIPS detectors. This calorimeter is made of a plastic scintillator (2 cm thick) that can detect electron fluxes up to 10 MeV. As a result, there is expected to be a reasonable overlap with the Zirè flux data, as noted in [21]. A bottom ACD (made of plastic scintillator) is also used to identify high-energy particles that are not completely contained by the plastic scintillator calorimeter.

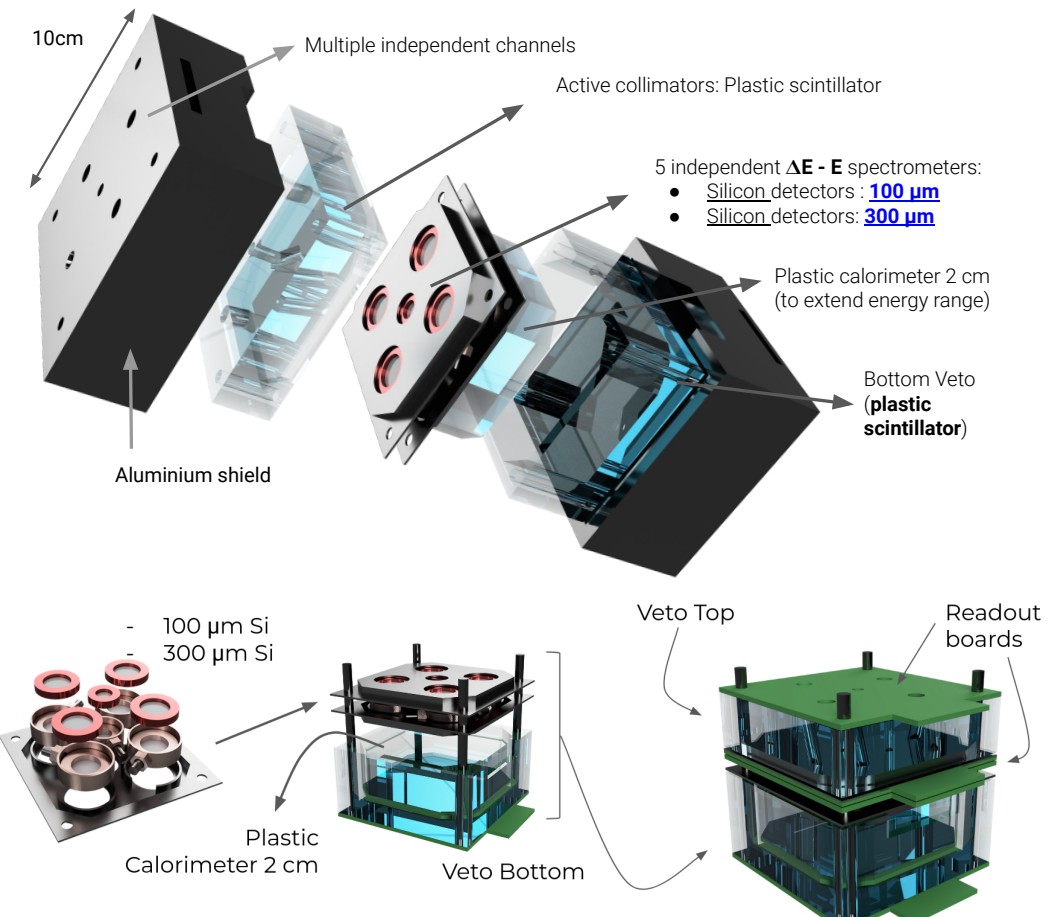

**Figure 2.** In the upper part of the figure, the expanded visualization of the LEM detector and its internal components. From the upper-left part of the figure: the aluminum shield has five holes/channels for the detector. The five channels are then evident in the active collimator, which is made of a plastic scintillator. In the core of the detector, the 5 Δ*E-E* detectors are positioned. After that, a calorimeter made of plastic scintillator is added to expand the energy range. In the lower part of the detector, there is an ACD made of plastic scintillator, followed by the bottom section of the aluminum shield. Below, the images describe the assembling and the geometry of the Low Energy Module (LEM) detector.

By considering a non-relativistic, low-energy charged particle traversing the Δ*E-E* telescope, the deposited energy in the thinner silicon detector, $\Delta E \propto \frac{Z^2}{\beta^2}$, and the particle's kinetic energy, $E_k \approx \frac{1}{2}m(\beta c)^2$, depends on the particle's velocity. If we combine these parameters, it is possible to define a particle classifier as

$$\text{PID} = \log_{10}\left(\frac{\Delta E}{1\,\text{MeV}} \cdot \frac{E_k}{1\,\text{MeV}}\right) \approx \log_{10}\left(Z^2 m\right) + \text{constant}. \tag{1}$$

This Particle IDentification (PID) classifier, dimensionless by definition, is primarily determined by the particle's mass, denoted as *m*, and its charge in modulus, represented by *Z*. As a result, this PID classifier partially (and approximately) does not depend on the velocity (and therefore on the kinetic energy) of the particle.

For the characterization of the detector's performances, a GEANT4 (version 11.0.3) Monte Carlo simulation [22] was appositely developed. The Physics List adopted in our application is the standard `FTFP_BERT`. For the simulation of the geometry reported in Figure 2, developed with a parametric computer-aided design (FreeCAD 0.20) software, we adopted the Geometry Description Markup Language (GDML) [23]. We generated the

GDML file (compatible with the GEANT4 toolkit) using the GDML Workbench [24] for FreeCAD 0.20. Figure 4 shows the event display generated by the GEANT4 application specifically developed for characterizing the detector. The cross-sectional view enables the reader to observe the tessellated solids used in the instruments.

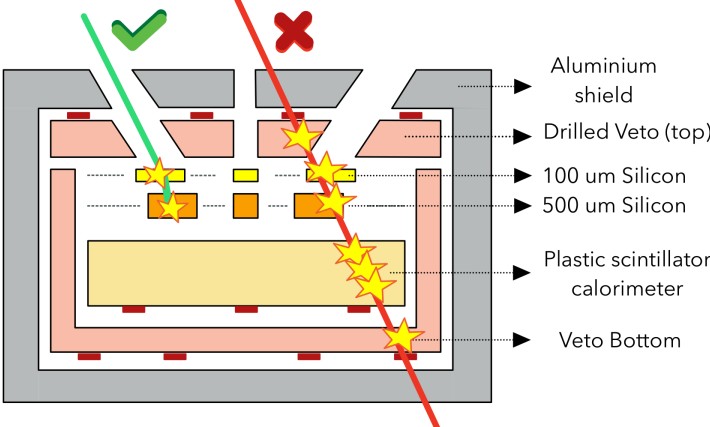

**Figure 3.** Schematics of the LEM detection approach; the green track is an example of good events (fully confined), and the red track represents an event to be rejected (not fully confined). The yellow star markers represent the energy deposited by the charged particle into the sensitive elements of the detector. Good events are characterized by a partial energy deposit in the thinner SD (100 μm) and a complete energy release in the thicker SD (300 μm) or, eventually, in the plastic calorimeter. In the second case, since the energy resolution of the plastic scintillator is worse, energy measurement will be affected by a larger uncertainty. Nevertheless, only when the energy release caused by the particle is confined within the detector an accurate PID is possible. Events to be rejected are characterized by an energy release in at least one of the two ACDs, or in more than two SDs not aligned on the same axis (e.g., two SDs that belong to different independent channels). Nonetheless, MIP particles (e.g., atmospheric muons on the ground), corresponding to crossing particles, will be used for calibration purposes.

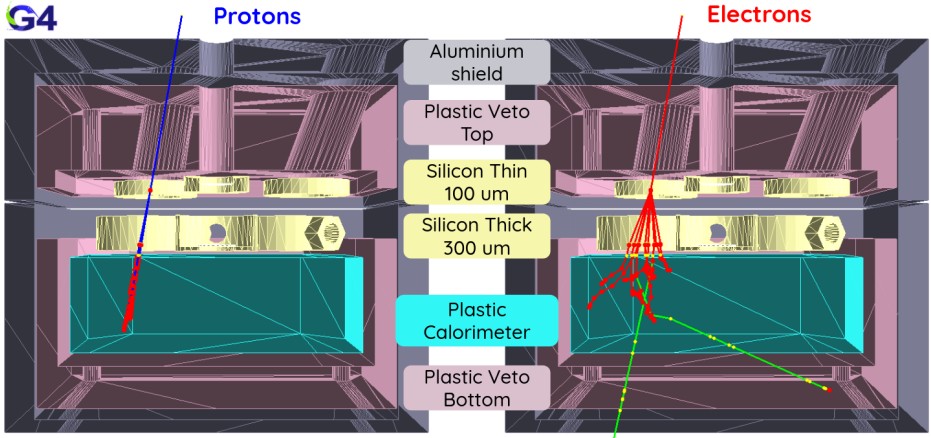

**Figure 4.** The picture shows the event display of the GEANT4 simulation of the Zire-LEM detector. (**Left Panel**) Visualization of 10-proton events (particle's trajectory is depicted by the blue line) with kinetic energy uniformly extracted between 3 and 50 MeV. (**Right Panel**) Visualization of 10-electron events (particle's trajectory is depicted by the red lines) with kinetic energy randomly extracted between 0.1 and 5 MeV. The green lines are photons produced during the electron's bremsstrahlung. Only in one case (displayed on the right-hand side of the left panel), the photon is re-absorbed via the photoelectric effect. It is possible to see that for electrons, the multiple scattering phenomenon is more impacting. This provides a graphical visualization of the need for an innovative active collimation technique for detecting the particle's direction at relatively low energy.

The PID classifier is shown in Figure 5a. It's interesting to note that the non-relativistic assumption is not valid for electrons. Nonetheless, they can still be identified since the electron's mass is ≈1/2000 times the proton's mass. However, the previously mentioned limited energy resolution for the plastic scintillator calorimeter can cause a reduction in PID capabilities and performance at relatively high energies, in particular when particles traverse the thicker silicon detector and deposit their energy in the plastic calorimeter.

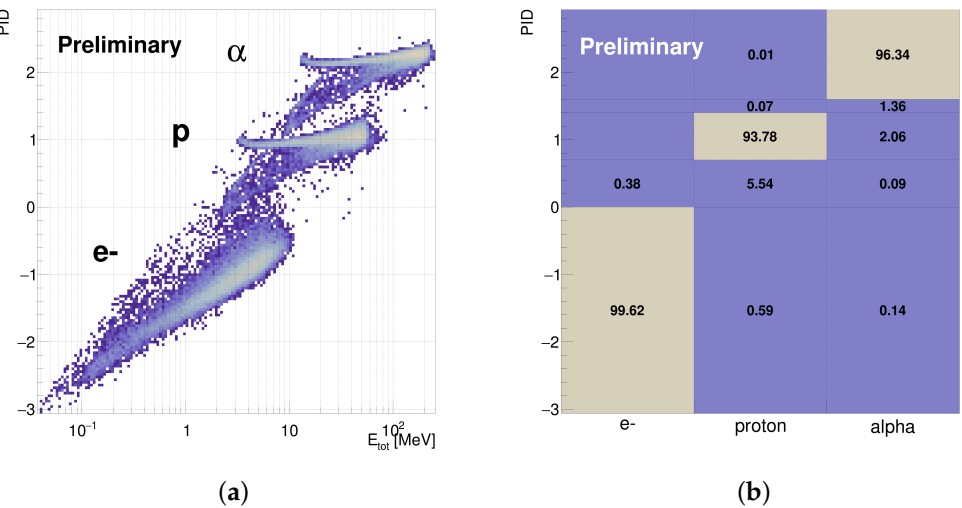

(a)                                                          (b)

**Figure 5.** (**a**) PID capability for events impinging on the top 100 μm silicon detector and fully contained within the LEM. (**b**) Particle tagging efficiency for the three families of particles: electrons, protons, and alpha particles. For each particle family (Monte Carlo truth) reported on the horizontal axis, the tagging efficiency is reported on each histogram bin.

To estimate the particle identification efficiency, it is possible to define some specific intervals for the PID for each particle: $(-3, 0)$ for electrons, $(0.7, 1.4)$ for protons, and $(1.6, 2.5)$ for alpha particles. In the table shown in Figure 5b, the particle identification tagging efficiencies, for the three families of particles, are higher than 90 % in the three respective PID proxy intervals. In particular, we observed that more than 90% of particles, for each of the three generated classes (electrons, protons, and alpha), were correctly tagged.

The angular resolution and FOV of the detector for protons and electrons are characterized in Figure 6. The scatter plot depicts a projection on the plane of the particle's incident direction (at the Monte Carlo truth level). The origin of the plot is assumed to represent the zenith direction, which is the axis perpendicular to the front drilled aluminum surface of the LEM. The color indicates which $\Delta E$-$E$ channel has been triggered. The entire LEM FoV is around 45°. The RMS angular resolution for protons and alpha particles is around 6°. We acquired a lower resolution (≈12°) for electrons. Interactions with the inner edges of the LEM openings on the top Aluminum shield were expected to cause such an observable effect. The LEM geometric factor is in the range 0.1–0.3 cm$^2$sr for electrons in the 0.2–12-MeV energy window, for protons in the 3–70 MeV energy window, and for alpha particles in the 15–280 MeV energy window. The estimation was carried out assuming the definitions and methods described in [25]. Figure 7 displays the estimated geometric factor for electrons, protons, and alpha particles. Knowing the orbit parameters of the NUSES mission (Sun-synchronous, 97 degrees, LEO 550 km), a preliminary map of the expected rates of the LEM can be evaluated using the model International Radiation Environment Near Earth AE9/AP9 (IRENE-AE9/AP9) [26]. In the LEO environment, the most impacting populations of charged particles are trapped protons and electrons. With IRENE-AE9/AP9 we could estimate the differential omnidirectional/isotropic fluxes of those particles.

Figure 8 shows that the LEM will encounter a significant acquisition rate (≈50 kHz) in the South Atlantic Anomaly (SAA). Therefore, a dual data transmission strategy is being developed. An "event-based" approach will be adopted for rates below 1 kHz. On the other hand, a "histogram-based" approach will be adopted for higher rates. This will ensure proper usage of the assigned data bandwidth to the LEM.

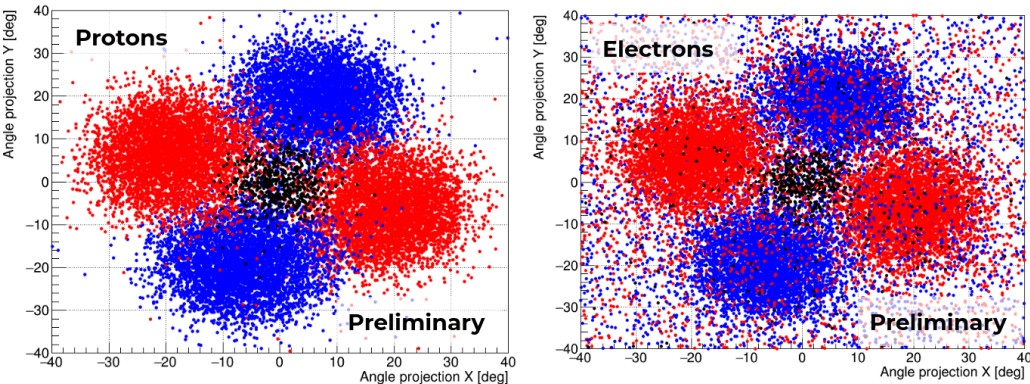

**Figure 6.** Angular resolution and Field Of View (FOV) of the LEM for protons on the left and electrons on the right. The different colors encode the pair in the $\Delta E$-$E$ spectrometer that is triggered. Since the detector exhibits axial symmetry, different colors are used to distinguish between adjacent lateral channels (in blue or red) and the central channel (in black). It is possible to see, on the right panel, the important effect of the electron's multiple scattering. Some electrons hitting the aluminum shield are then scattered in the direction encoded by one of the five channels.

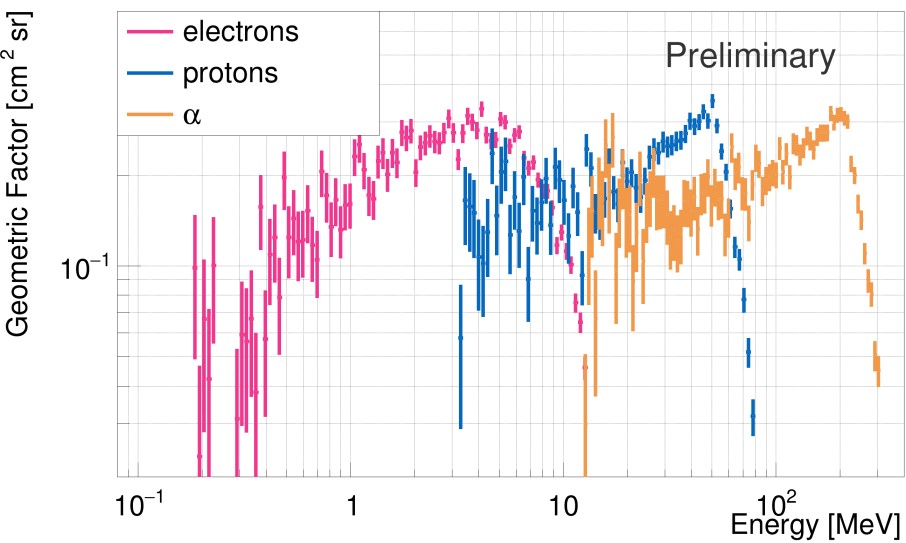

**Figure 7.** Geometric factor estimation for the LEM.

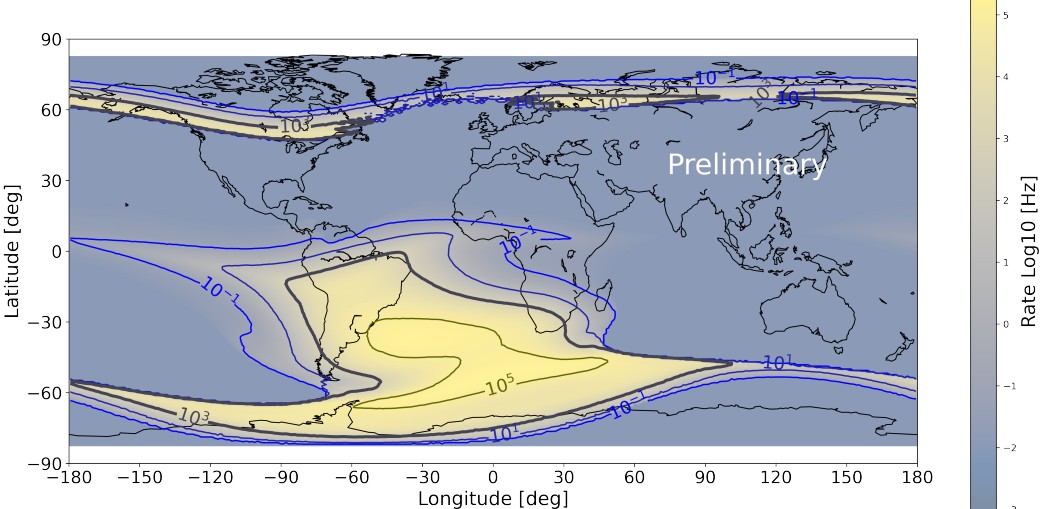

**Figure 8.** Estimated rate map having considered a satellite polar orbit, Sun-synchronous 97° and 550 km of altitude. For the conversion from isotropic fluxes to rates we used an $\simeq 0.2$ cm²sr geometric factor. The thicker contour in the map represents the region inside which the LEM will operate in the histogram-based mode.

## 4. Preliminary Test on PIPS Sensors

The core of the LEM detector is constituted by the $\Delta E$-$E$ spectrometers, which comprise five detectors. Four of these pairs of detectors have a circular shape with an area of 150 mm². The central pair of detectors has an area of 55 mm² (as depicted in Figure 9). The smaller diameter of the central PIPS detector was chosen to ensure consistent geometric acceptance across all five channels. The top sensors, each with a thickness of 100 μm, will be the R-series (ruggedized) PIPS detectors produced by ORTEC/AMETEK [27].

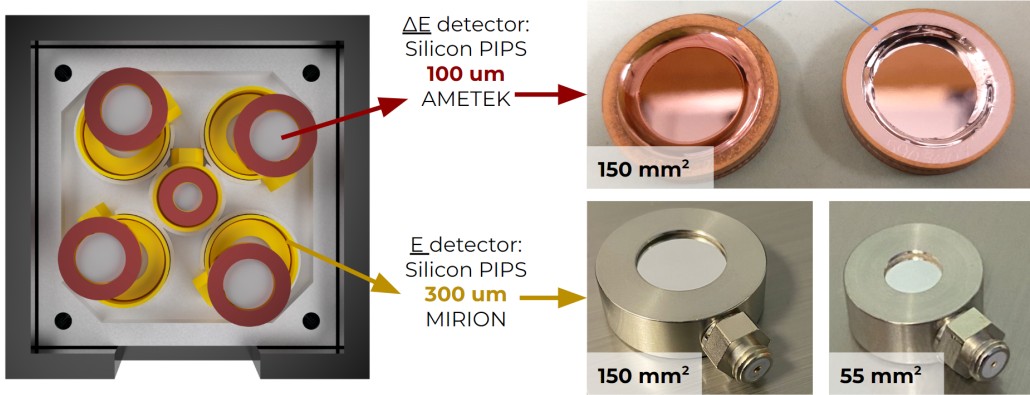

**Figure 9.** Mounting arrangement of the 5 pairs of silicon detectors within the LEM. On the right-hand side of the picture, some pictures of the PIPS detectors manufactured by AMETEK/ORTEC and by MIRION/CANBERRA are reported.

The 100 μm PIPS detectors are covered by aluminum and gold layers on both sides. Each layer has a grammage of 50 μg/cm² and 40 μg/cm², respectively. These layers are important for making the PIPS detectors light-tight and "ruggedized". On the other hand, the five bottom sensors, with a thickness of 300 μm, will be produced by Canberra/MIRION [28]. The 300 μm thick PIPS detectors have aluminum layers with a grammage of approximately 70 μg/cm² and 250 μg/cm² on their two sides. This treatment ensures that the detectors are also light-tight.

An experimental setup to assess the performance of a PIPS detector was developed at the INFN-TIFPA laboratory. During the characterizations, the PIPS sensor's depletion

voltage was set at 60 V. An initial measurement of the power consumption of the employed Charge-Sensitive Preamplifier (CSA) was below 100 mW per channel. We characterized the sensor acquiring various particle types in a telescopic configuration, including atmospheric muons, $\gamma$-rays from a $^{176}$Lu source, as well as alpha particles and $\gamma$-rays from a $^{241}$Am source. During these tests, a good linearity of the energy scale was achieved. Furthermore, the PIPS detector's response to particles with significantly different specific ionization levels (such as muons, recoiling electrons, and alpha particles) was determined to be consistent within a few percentage points, as anticipated.

In Figure 10 are reported some preliminary results from the characterization of the DAQ developed by Nuclear Instruments SRL. For the test, the silicon detector AP-CAM25 (manufactured by Mirion with an embedded CSA) was used. The detector was exposed to different radioactive sources: $^{137}$Cs and $^{241}$Am. $^{137}$Cs, with a branching ratio of 94% decays in an excited state of $^{137}$Ba through a beta decay. The excited state relaxes releasing a 662 keV photon. Since Silicon is a material with a low atomic number, the photoelectric peak is suppressed. Nevertheless, it is possible to record the Compton edge of the spectrum with an end-point at $\approx$478 keV. Moreover, the detector was exposed to the 59.5-keV gamma line emission by the $^{241}$Am source.

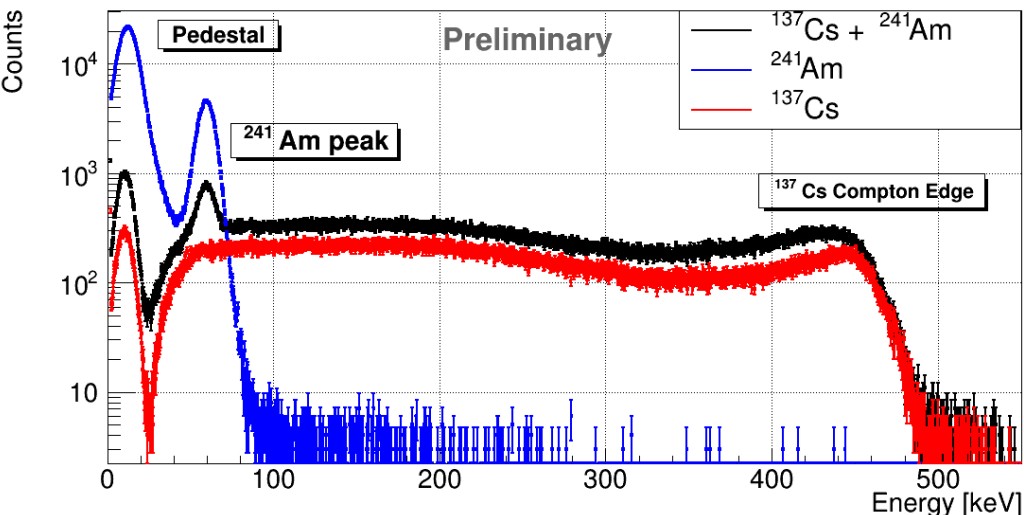

**Figure 10.** Some preliminary measurements were performed using the DAQ manufactured by Nuclear Instruments SRL. For these measurements, a fully depleted silicon detector with an embedded CSA was used (AP-CAM25 manufactured by MIRION). The spectra were acquired by exposing the detector to different radioactive sources. As indicated in the legend, the red curve represents the Compton edge of the 662 keV gamma-ray from the decay of $^{137}$Cs. The blue spectrum displays the peak at 59.5 keV from the decay of the $^{241}$Am radioactive source. The black spectrum was obtained by exposing the silicon detector to both radioactive sources, $^{241}$Am and $^{137}$Cs.

There are two important requirements for the readout chain of the LEM: a relatively low energy threshold and a rapid response. The measurements (Figure 10) obtained confirmed the practicability of the 40 keV energy threshold utilized in LEM simulations and validate the achievement of a $\approx$5-keV energy resolution. Moreover, another crucial aspect of the LEM concerns the characterization of the signal decay time (defined by the time constant of the CSA), which is connected to the detector occupancy—a possible concern in the harsh conditions of the South Atlantic Anomaly (SAA). The measured signal decay time of approximately 200 ns effectively mitigates the risk of signal overlap (pile-up) from various particles in the SAA, thus addressing this potential issue.

## 5. Conclusions

The Low-Energy Module (LEM), a small particle analyzer that will be part of Zirè on-board the NUSES mission, is currently being built and tested. The INFN-TIFPA laboratory

tested prototypes of the PIPS detector readout, confirming an energy resolution of around 5 keV and a signal decay time of around 200 ns. The LEM is intended to monitor the energy, direction, and composition of relatively low-energy charged particles, with a kinetic energy limit of 0.1 MeV. Its goals include measuring particle fluxes in the South Atlantic Anomaly (SAA), studying the interaction of the lithosphere and magnetosphere, and monitoring Space Weather.

**Supplementary Materials:** The following supporting information can be downloaded at: https://www.mdpi.com/article/10.3390/instruments7040040/s1, for detailed author affiliations please see File S1. Full NUSES Author List: R. Aloisio, C. Altomare, F.C.T. Barbato, R. Battiston, M. Bertaina, E. Bissaldi, D. Boncioli, L. Burmistrov, I. Cagnoli, M. Casolino, A.L. Cummings, N. D'Ambrosio, I. De Mitri, G. De Robertis, C. De Santis, A. Di Giovanni, A. Di Salvo, M. Di Santo, L. Di Venere, J. Eser, M. Fernandez Alonso, G. Fontanella, P. Fusco, S. Garbolino, F. Gargano, R.A. Giampaolo, M. Giliberti, F. Guarino, M. Hellero, R. Iuppa, J.F. Krizmanic, A. Lega, F. Licciulli, F. Loparco, L. Lorusso, M. Mariotti, O.M.N. Mazziotta, M. Mese, H. Miyamoto, T. Montarulio, A. Nagaio, R. Nicolaidis, F. Nozzoli, A.V. Olinto, D. Orlandi, G. Osteria, P.A. Palmieri, B. Panico, G. Panzarini, A. Parenti, L. Perrone, P. Picozza, R. Pillera, R. Rando, O.M. Rinaldi, A. Rivetti, V. Rizi, F. Salamida, E. Santero Mormile, V. Scherini, V. Scotti, D. Serini, I. Siddique, L. Silveri, A. Smirnov, R. Sparvoli, S. Tedesco, C. Trimarellio, L. Wu, P. Zuccon, S.C. Zugravel.

**Author Contributions:** Conceptualization, F.N. and R.N.; methodology, F.N. and R.N.; software, R.N.; validation, F.N. and R.N.; formal analysis, R.N.; investigation, F.N. and R.N.; resources, F.N., R.N., G.P. and on behalf of the NUSES coll.; data curation, R.N. and F.N.; writing—original draft preparation, R.N. and F.N.; writing—review & editing, R.N., F.N., G.P. and on behalf of NUSES coll.; visualization, R.N., F.N. and G.P.; supervision, F.N. and G.P.; project administration, F.N., G.P. and NUSES coll.; and funding acquisition, F.N., G.P. and on behalf the of NUSES coll. All authors have read and agreed to the published version of the manuscript.

**Funding:** NUSES is funded by the Italian Government (CIPE n. 20/2019), by the Italian Minister of Economic Development (MISE reg. CC n. 769/2020), by the Italian Space Agency (CDA ASI n. 15/2022), by the Swiss National Foundation (SNF grant n. 178918) and by the European Union-NextGenerationEU under the MUR National Innovation Ecosystem grant ECS00000041-VITALITY-CUP D13C21000430001.

**Institutional Review Board Statement:** Not applicable.

**Informed Consent Statement:** Not applicable.

**Data Availability Statement:** The data presented in this study are available on request from the corresponding author.

**Conflicts of Interest:** The authors declare no conflict of interest.

## Abbreviations

The following abbreviations are used in this manuscript:

| | |
|---|---|
| ACD | Anti-Coincidence Detector |
| CSA | Charge Sensitive Amplifier |
| EAS | Extensive Air Showers |
| FOV | Field Of View |
| GDML | Geometry Description Markup Language |
| GEANT4 | GEometry ANd Tracking 4 |
| GRB | Gamma-Ray Burst |
| IRENE | International Radiation Environment Near Earth |
| LAIM | Lithosphere Atmosphere Ionosphere Magnetosphere |
| LEM | Low-Energy Module |
| LEO | Low Earth Orbit |
| LYSO | Lutetium–Yttrium OxyorthoSilicate |
| MILC | Magnetosphere ionosphere lithosphere coupling |

| MIP | Minimum Ionizing Particle |
| MPV | Most Probable Value |
| NUSES | NeUtrino and Seismic Electromagnetic Signals |
| PID | Particle Identification |
| PIPS | Passivated Implanted Planar Silicon |
| SAA | South Atlantic Anomaly |
| TGF | Terrestrial Gamma-ray Flash |
| UAS | Upgoing Air Showers |
| VAB | Van Allen Belt |

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
