# Peer review of "A Compact Particle Detector for Space-Based Applications: Development of a Low-Energy Module (LEM) for the NUSES Space Mission"

_instruments, doi:10.3390/instruments7040040_

Round 1
Reviewer 1 Report
Comments and Suggestions for Authors
This work presents the development of the Low Energy Module for the NUSES experiment.
The scientific motivation of this experiment is clearly described along with the detector set-up. The paper is very well-written, concise and informs the reader about the status of the LEM detector development.
My only crucial concern relates to section 4:
Between lines 173-191, a series of very interesting and important characterisation measurements are mentioned. Nevertheless, those results are not supported by some plots that will convince the reader about their correctness. On this, I strongly suggest adding some results or reducing their importance in this paper and presenting them in a future publication.
As there are no methodology issues that will require further data analysis and supposing that the results are ready and a small amount of time will be required to include them in the paper, I will suggest a minor revision.
I also request the authors to make the following minor corrections in the text:
Figure 1 caption: "Its goal is to keep incredibly low-energy particles" What is this energy? It would be helpful for the reader to quantify this.
Figures 2 and 3: Replace u with the Greek letter μ for microns.
Figure 2 (caption, 3rd line): Active collimator -> active collimator
Lines 68 and 97: fig -> Fig
Line 115: Double dot after const
Line 132: I guess the PID proxy parameter is unit-less, if so please clarify in line 115, otherwise the reader gets confused in figure 5.
Line 136: what are the errors of this estimation? By providing the error the reader is informed if your simulation was adequate.
Lines 139-142: Correct Latex issues (Delta, alpha, approx)
All text: Between numbers and units sometimes there is a space and sometimes not, please make the text uniform.
Figure 6: "The color indicates which ΔE-E channel has been triggered", but the channel information is not provided. Please add this information to the text or with a legend in the plot.
References:
The style is not homogeneous, please work on this.
Ref 1,2 and 9 now are published, please provide publication details and DOI instead of arXiv preprint
Ref 4 and 5 are incomplete.
I strongly suggest adding a DOI link to all the references
Reviewer 2 Report
Comments and Suggestions for Authors
Dear Authors,
I have reviewed your study on the NUSES space mission, which explores observational and technological methods for cosmic rays, gamma rays, and neutrinos. The Terzina and Zirè payloads introduce innovative approaches across different energy ranges.
Recognizing the importance of these findings in cosmic ray measurements, I recommend publication with the minor revisions I've outlined below. You can find my specific feedback in the following sections.
Best regards,
Reviewer
General comments:
1. On line 103 and in other instances throughout the paper, please refrain from italicizing the 'mu' symbol. It should be italicized when representing a variable, but not when it's part of a unit. If you are using LaTeX, consider implementing the '\usepackage{upgreek}' package, which ensures that Greek letters are not italicized when used this way.
2. Kindly include 'NUSES' in the list of abbreviations.
3. It would be beneficial for readers who may not have specialized knowledge to clarify that when discussing low-energy cosmic rays, the term "low-energy" applies to electrons in the 0.1-7 MeV range and protons in the 3-50 MeV range. In certain fields, electron and proton energies above 1 MeV are considered high-energy. This clarification would enhance understanding.
4. Line 15. The phrase is generally correct, but it could be written more clearly. Here's sugested (could be ignored) a revised version: "...in which the anticipated electron fluxes are on the order of 10^6 to 10^7 electrons per square centimeter per steradian per second."
5. Line 47 and in other instances through the paper. The academically correct format would be "10x10x10 cm³ volume," where the "cm³" (cubic centimeters) unit is properly spaced from the preceding number and separated by a space from the subsequent text. This format follows standard scientific notation guidelines.
6. It would be helpful to specify that "100 µm" and "100 um" (as seen in Figure 3) are equivalent for the reader's clarity.
7. The statement "These authors contributed equally to this work" doesn't align with the details provided in the "Author Contributions" section.
Specific comments:
1. Figure 1: Providing the outer dimensions of the NUSES platform for general readers would be helpful. Additionally, including approximate weight information for both NUSES and LEM modules would be beneficial.
2. Line 38. In this context, should the word 'broadcasts' be changed to 'forecast'? If not, could you kindly provide more details regarding the concept of space weather broadcasting?
3. Line 97 - 98. References to figures in the text sometimes start with a capital letter and sometimes with a small letter. Please use the format suggested in the journal template for consistency.
4. I noticed there's no reference to Figure 4 in the text. The Geant4 Monte Carlo simulation you presented is interesting and significant. If possible, could you provide the Geant4 source files and analysis scripts for readers to reproduce and verify the results? This would be a valuable Geant4 example for cosmic ray measurements. Additionally, specifying the Geant4 version and data libraries used would be helpful.
5. Line 140. Could you clarify how the zenith direction is defined? I've noticed that sometimes authors use 'DeltaE,' and in other cases, 'Delta' is indicated with the Greek letter. Consistency in the notation would be helpful.
6. Line 141 and line 146. I've noticed that sometimes authors use 'alpha' and in other cases, 'alpha' is indicated with the Greek letter. Consistency in the notation would be helpful.
7. Line 142. Would it be more consistent to replace 'approx' with the approximate sign?
8. Preliminary test on PIPS sensors section. Is it possible to estimate the life expectancy of the LEM detector when exposed to low-energy cosmic rays?
9. Preliminary test on PIPS sensors section. Could you please provide a rough estimate of the funds needed to construct the LEM detector?
